# Novel ABC Transporter Associated with Fluconazole Resistance in Aging of *Cryptococcus neoformans*

**DOI:** 10.3390/jof8070677

**Published:** 2022-06-28

**Authors:** Natalia Kronbauer Oliveira, Somanon Bhattacharya, Rina Gambhir, Manav Joshi, Bettina C. Fries

**Affiliations:** 1Department of Microbiology and Immunology, Renaissance School of Medicine, Stony Brook University, Stony Brook, NY 11794, USA; natalia.kronbauerdeoliveir@stonybrook.edu; 2Division of Infectious Diseases, Department of Medicine, Stony Brook University, Stony Brook, NY 11794, USA; somanon.bhattacharya@stonybrookmedicine.edu; 3Department of Medicine, Stony Brook University, Stony Brook, NY 11794, USA; rina.gambhir@stonybrook.edu (R.G.); manav.joshi@stonybrook.edu (M.J.); 4Veterans Administration Medical Center, Northport, NY 11768, USA

**Keywords:** drug resistance, ABC transporter, efflux pump, cryptococcosis, aging

## Abstract

*Cryptococcus neoformans* causes meningoencephalitis in immunocompromised individuals, which is treated with fluconazole (FLC) monotherapy when resources are limited. This can lead to azole resistance, which can be mediated by overexpression of ABC transporters, a class of efflux pumps. ABC pump-mediated efflux of FLC is also augmented in 10-generation old *C. neoformans* cells. Here, we describe a new ABC transporter Afr3 (*CNAG_06909*), which is overexpressed in *C. neoformans* cells of advanced generational age that accumulate during chronic infection. The Δ*afr3* mutant strain showed higher FLC susceptibility by FLC E-Test strip testing and also by a killing test that measured survival after 3 h FLC exposure. Furthermore, Δ*afr3* cells exhibited lower Rhodamine 6G efflux compared to the H99 wild-type cells. Afr3 was expressed in the *Saccharomyces cerevisiae* ADΔ strain, which lacks several drug transporters, thus reducing background transport. The ADΔ + Afr3 strain demonstrated a higher efflux with both Rhodamine 6G and Nile red, and a higher FLC resistance. Afr3-GFP localized in the plasma membrane of the ADΔ + Afr3 strain, further highlighting its importance as an efflux pump. Characterization of the Δ*afr3* mutant revealed unattenuated growth but a prolongation (29%) of the replicative life span. In addition, Δ*afr3* exhibited decreased resistance to macrophage killing and attenuated virulence in the *Galleria mellonella* infection model. In summary, our data indicate that a novel ABC pump Afr3, which is upregulated in *C. neoformans* cells of advanced age, may contribute to their enhanced FLC tolerance, by promoting drug efflux. Lastly, its role in macrophage resistance may also contribute to the selection of older *C. neoformans* cells during chronic infection.

## 1. Introduction

*Cryptococcus neoformans* is an opportunistic yeast that infects immunocompromised individuals, causing meningoencephalitis. The most recent global data estimates that cryptococcosis affects approximately 223,100 people annually, resulting in 181,100 fatalities. Globally, this invasive fungal infection is the cause of 15% of all AIDS-related deaths [1]. The standard treatment includes amphotericin B (AMB) and 5-fluorocytosine (5-FC) as induction therapy, followed by prolonged treatment with fluconazole (FLC) for maintenance therapy [2]. In countries with limited resources, however, FLC monotherapy is used as an alternative treatment. High mortality, treatment failure, and fluconazole resistance have been described in association with FLC monotherapy [3,4,5].

FLC is a triazole that inhibits lanosterol 14α-demethylase, an enzyme encoded by the *ERG11* gene, which is a rate-limiting step for ergosterol biosynthesis [6,7]. Azole resistance emerges through different mechanisms. Overexpression and mutations in *ERG11* decrease the susceptibility to azoles, while overexpression of efflux pumps decreases intracellular drug concentration [8,9,10,11]. The efflux of azoles is facilitated by the ATP-binding cassette (ABC) transporters, a class of pumps that use ATP as an energy source to drive transport [12]. Three efflux pumps have been characterized in *C. neoformans*. Afr1 is the primary azole efflux pump and deletion of the *AFR1* gene leads to lower FLC minimum inhibitory concentrations (MICs), while its overexpression causes increased FLC resistance [13]. Single deletions of another two ABC transporters, Afr2 and Mdr1, cause no change in FLC MICs. Deletion of all three transporters (Afr1, Afr2, and Mdr1) increases susceptibility to FLC compared to Afr1, suggesting that Afr1 acts as the major efflux pump, while Afr2 and Mdr1 augment Afr1 function [14]. Of note is that, when these ABC transporters were expressed in *Saccharomyces cerevisiae*, increased resistance to multiple azoles was observed. Furthermore, a *S. cerevisiae* mutant that expresses these *C. neoformans* transporters showed lower accumulation of radiolabeled FLC [11]. Additionally, a previous study demonstrated increased expression of these transporters in response to FLC, indicating that expression of Afr1 and Afr2 was induced by FLC [14]. A search in the *Cryptococcus* genome database revealed a total of 41 ABC transporters in *C. neoformans.* Most transporters are not characterized, and it is not understood if their expression contributes to FLC resistance in *C. neoformans*.

Replicative aging is a conserved trait of eukaryotic organisms and in *C. neoformans*, it is linked closely with resilience in the host environment and increased FLC tolerance in 10-generation-old cells [15,16]. Replicative aging is the result of asymmetric cell divisions. In the course of these divisions, the mother cells progressively age and accumulate age-associated phenotypic changes. Interestingly, in *S. cerevisiae*, ABC transporters are asymmetrically distributed in the course of replicative aging [17]. Aged mother cells exhibit increased resistance to phagocytosis and macrophage-mediated killing [15,18], as well as increased tolerance to anti-fungal drugs, including FLC. A cell can undergo a finite number of divisions before senescence and the total number of divisions constitutes their replicative life span (RLS) [19], which varies greatly among clinical *C. neoformans* strains.

Transcriptome analysis of young and old *C. neoformans* cells has demonstrated differential expression of the gene *CNAG_06909*. This gene was identified based on homology to encode an ABC transporter. Expression was 16-fold upregulated in 10-generation old *C. neoformans* cells, compared to young 0–3 generation-old cells. Due to similarities of the *CNAG_06909* efflux pump to the previously described ABC transporters Afr1 and Afr2, we renamed the *CNAG_06909* protein Afr3.

Here, we investigated the role of Afr3 in virulence and FLC resistance. We investigated virulence and FLC sensitivity of an Δ*afr3* mutant strain and measured efflux function. To exclude compensation by other efflux pumps, we expressed Afr3 in the *S. cerevisiae* strain (ADΔ), which lacks other ABC transporters [6]. The present study demonstrates that Afr3 is an ABC transporter that is upregulated in old cells where it effluxes FLC, which may contribute to the age-dependent FLC tolerance in *C. neoformans*.

## 2. Materials and Methods

### 2.1. Strains and Media

*C. neoformans* strains H99 and Δ*afr3*, Δ*afr1*, and Δ*afr2* mutants were cultured in synthetic media (SM; 1.7 g yeast nitrogen base without amino acids (BD, Frankin Lakes, NJ, USA), 1 g drop out mix (USBiological Life Sciences, Salem, MA, USA), 4 mL ethanol, 5 g (NH_4_)_2_SO_4_, 3.3 g NaCl, 20 g glucose). Calorie restriction synthetic media was prepared as follows: CR, 1.7 g yeast nitrogen base without amino acids (BD, Frankin Lakes, NJ, USA), 1 g drop out mix (USBiological Life Sciences, Salem, MA, USA), 4 mL ethanol, 5 g (NH_4_)_2_SO_4_, 3.3 g NaCl, 0.5 g glucose. The Δ*afr3* strain was derived from the Madhani knockout collection, which is managed by the Fungal Genetics Stock Center. The Δ*afr1* and Δ*afr2* strains were gifted by Dr. Kwon-Chung Lab [14]. The mutant *S. cerevisiae* strains were cultured in complete supplement medium without uracil (CSM-Ura (MP Bio, Irvine, CA, USA); 20 g galactose, 1.7 g YNB, 5 g (NH_4_)_2_SO_4_, 0.77 g CSM-Ura). *S. cerevisiae* ADΔ strain, in which several ABC transporters and the *URA3* locus were deleted, was previously described [20]. The ADΔ strain and pYES2 plasmid were obtained as a gift from Dr. Theodore C. White at the University of Missouri, Kansas City. All strains used in this study are maintained as 30% glycerol stocks and stored at −80 °C for future use.

### 2.2. Construction of S. cerevisiae Strain Expressing Efflux Pumps

First, RNA was extracted from exponentially growing H99 cells using the RNAeasy Plus kit (Qiagen, Hilden, Germany), following the manufacturer’s guidelines. Next, 250 ng of RNA was converted to cDNA using the Verso cDNA Kit (Thermo Fisher Scientific, Waltham, MA, USA) in a 20 μL reaction. *AFR3* was amplified from the cDNA using oligos that partially overlapped with the pYES2 plasmid (Appendix A), using a thermocycler (Biorad, Hercules, CA, USA). *AFR3* was amplified from cDNA instead of gDNA, due to the presence of introns that could lead to aberrant splicing. The forward oligonucleotide was designed to contain 40 bp homologous sequence of *GAL1* promoter (primer *AFR3* + p*GAL1* F), while the reverse oligonucleotide was designed to contain 40 bp homologous sequence of *CYC1* terminator (primer *AFR3* + *CYC1* tt R). The plasmid contains a *URA3* auxotrophic selection marker. pYES2 was first digested with HindIII and then transformed into ADΔ strain with the addition of the *AFR3* cassette, as previously described [21]. The ADΔ strain lacks seven ABC transporters (Δ*yor1*, Δ*snq2*, Δ*pdr10*, Δ*pdr11*, Δ*ycf1*, Δ*pdr5*, and Δ*pdr15*), a transcription factor (Δ*pdr3*), and the *URA3* gene (Δ*ura3*). Homologous recombination was used to integrate the *AFR3* cassette into pYES2. Undigested plasmid pYES2 without the cassette was transformed into ADΔ for positive control. Transformants were selected on CSM-Ura agar plates after incubation at 30 °C for 4 days. To screen for proper integration, 12–15 colonies were selected and replicated into fresh selective plates three consecutive times. Transformants were confirmed by plasmid extraction with the QIAprep Spin Miniprep Kit (Qiagen, Hilden, Germany) and PCR for the *AFR3* cassette, using oligonucleotides that amplified the whole gene sequence (primers *AFR3* F and *AFR3* R) (Appendix A). Expression was measured using the RNAeasy Plus kit (Qiagen, Hilden, Germany) to extract RNA from ADΔ and ADΔ + Afr3 and RNA was converted to cDNA using the Verso cDNA Kit (Thermo Fisher Scientific, Waltham, MA, USA). cDNA was diluted 1:5 and analyzed with qPCR analysis (Roche, Basel, Switzerland) using the Power Sybr Green Master Mix (Applied Biosystems, Waltham, MA, USA) following the manufacturer’s protocols (primers qPCR Afr3 F and qPCR Afr3 R, Appendix A). The housekeeping gene encoding β-actin (primers Sc *ACT1* F and Sc *ACT1* R) was used as an internal control (Appendix A). Furthermore, samples were sent for sequencing to ensure that the *AFR3* sequence did not undergo mutations.

Construction of the catalytically inactive Afr3 mutant lacking the nucleotide-binding domain (NBD) was performed by amplifying *AFR3* from H99 cDNA without the first 835 bp (primers NBD + p*GAL1* F and AFR3 + *CYC1* tt R) (Appendix A). Transformation and confirmation of transformants (Appendix A) were performed as described above, employing the same primers used for construction. Afr3-GFP construction for the cellular localization analysis proceeded in two stages. First, the GFP sequence was amplified from the pFA-6A-GFP plasmid (Addgene, Watertown, MA, USA; primers GFP F and GFP R), employing oligonucleotides that partially overlapped with the pYES2 plasmid *GAL1* promoter and *CYC1* terminator. A Hind III restriction site was also added at the 5′ end of GFP. Transformation and transformants selection were performed as described above. The pYES2-GFP plasmid was extracted from ADΔ cells and digested with Hind III. *AFR3* was amplified from H99 cDNA (primers *AFR3* + p*GAL1* F and *AFR3* GFP R), using oligonucleotides that partially overlapped with the *GAL1* promoter and the GFP sequence. The oligonucleotides were designed in such a way that the Afr3 stop codon was removed. Transformation, selection, and confirmation were performed as described above (primers *AFR3* F and GFP R) (Appendix A).

### 2.3. Isolation of Old C. neoformans Strains

Isolation of 10-generation-old *C. neoformans* cells was performed following the previously published protocol [22]. Briefly, the cells from the strains H99, Δ*afr1*, and Δ*afr3* were incubated overnight at 37 °C in SM media. The next day, the overnight cultures were washed 3 times with 1× PBS and were diluted 1:50 times. The diluted cells were then exponentially grown for 6–8 h. After the exponential growth, the cells were washed 3 times with 1× PBS and counted with a hemocytometer. Then, 10^8^ cells from each strain were labeled with 8 mg/mL sulfosuccinimidyl-6-[biotin-amido] hexanoate (Sulfo-NHS-LC-LC-Biotin, Thermo Fisher Scientific, Waltham, MA, USA) for 30 min at room temperature (RT). The labeled cells were then washed 3 times in 1× PBS and the washed cells were grown in fresh SM media for 5 generations (12–15 h). After 5- generation cell growth for each strain, the cells were washed 3 times with 1× PBS and labeled with 100 µL of streptavidin microbeads (Miltenyi Biotec, Bergisch Gladbach, Germany) at a final concentration of 10^8^ cells/mL. The streptavidin labeling was carried out for 15 min at 4 °C. The labeled cells were washed 3 times with 1× PBS to remove any unbound streptavidin. The biotin-streptavidin labeled 5 generation cells were then separated by passing the mixed population through AutoMACS*^®^* Pro Separator (Miltenyi Biotec, Bergisch Gladbach, Germany). The positively labeled cells were retained in the column attached to the magnet in the pro-separator machine. These cells were retrieved once the magnetic field was removed. These 5-generation old cells were then further grown in fresh SM media for another 5 generations (12–15 h). The grown cells were again washed and passed through AutoMACS*^®^* Pro Separator (Miltenyi Biotec, Bergisch Gladbach, Germany) to retrieve the 10 generation old cells. The purity of the population was verified by microscopy. As a control, the young generation cells, washed off from the magnetic columns, from the second separation were used.

### 2.4. Antifungal Susceptibility Testing

The minimum inhibitory concentration (MIC) was determined as per a previously published protocol [23]. Briefly, the *C. neoformans* strains were cultured overnight at 37 °C, and cells were adjusted to 10^5^ cells/well. FLC was 2-fold serially diluted in a flat-bottom 96-well plate (Corning Costar, Corning, NY, USA), with a starting FLC concentration of 64 µg/mL. The plates were incubated at 37 °C for 4 days and the OD_600_ was measured (SpectraMax i3x, Molecular Devices, San Jose, CA, USA). A row with no drugs was used as a growth control, while a row with no cells was used as a contamination control in the MIC assay. MICs were defined as the minimum drug concentration that inhibits 80% of the cell growth (MIC_80_). The assay was performed in triplicate. For analysis of MICs under CR, the same conditions were performed, in which the cells were cultured in synthetic media with 0.05% glucose. FLC E-Test was also performed to determine FLC susceptibility. Then, 10^6^ *C. neoformans* and 10^7^ *S. cerevisiae* cells were plated in YPD media for *C. neoformans* and CSM -Ura media with 2% galactose for *S. cerevisiae* containing the FLC E-Test strip (bioMerieux, Marcy-l’Etoile, France) and incubated at 37 °C or 30 °C for 4 days.

FLC killing assay was performed using the previously published protocol [24]. Briefly, the young and the 10 generation-old cells were isolated as described above. After isolation, the cells were washed 3 times with 1× PBS and 10^4^ cells per well were seeded in 96 well plates (Corning Costar, Corning, NY, USA) containing FLC at concentrations of 50, 25, 12.5, 6.25, and 3.125 µg/mL. Cells were also plated in wells containing no drug. Next, the 96 well plates were incubated at 37 °C for 3 h without shaking. After incubation, the cells were diluted 50 times and plated in YPD agar plates. The agar plates were incubated for 48 h and the colony-forming units (*CFUs*) were counted. The assay was performed in triplicate. Percent killing was analyzed by the following formula:$$\%\;\mathrm{killing}=\frac{CFUs\;in\;no\;drug\;well-CFUs\;in\;drug\;well}{CFUs\;in\;no\;drug\;well}\times 100\;$$

### 2.5. Rhodamine 6G Efflux Assay

Rhodamine 6G assay was performed as previously described [25]. Briefly, 5 × 10^7^
*C. neoformans* and 3 × 10^7^
*S. cerevisiae* cells were starved for 2 h in a phosphate buffer saline (PBS; pH 7.4) buffer at room temperature. Rhodamine 6G (Sigma-Aldrich, St. Louis, MO, USA) was added to a final concentration of 10 μM and the cells were incubated at 37 °C or 30 °C for 30 min. Following the incubation, cells were washed 3× in PBS, and efflux was initiated by the addition of 2% glucose. Samples were collected at 0 min, 10 min, 20 min, and 30 min timepoints, and fluorescence of the supernatants was measured at 525 nm excitation and 555 nm emission wavelengths. The experiment was performed independently on three different days.

### 2.6. Nile Red Assay

Briefly, the cells from ADΔ and ADΔ + Afr3 were grown in the CSM-Ura media. After growth, 10^7^ cells were used for the Nile Red assay. First, the cells were washed in PBS 3 times and then starved for 2 h in PBS to get rid of any residual glucose. After starvation, Nile red was added to the cells at a final concentration of 7 µM. The efflux was initiated after the addition of 2% glucose to the starved cells. Accumulation of Nile red was measured using a fluorescence plate reader (SpectraMax I3, Molecular Devices, San Jose, CA, USA) at 0 min and 30 min using the excitation wavelength of 553 nm and emission of 636 nm. Accumulation was calculated in percentage. More accumulation of the dye at the end of 30 min period signifies lesser efflux. The assay was conducted in triplicate.

### 2.7. Cellular Localization

Cellular localization of Afr3 was determined utilizing the Afr3-GFP construct transformed into the *S. cerevisiae* ADΔ strain and an ADΔ strain transformed with the empty pYES2 plasmid as control. Cells were grown overnight in CSM-Ura media containing 2% galactose and fixed with 4% paraformaldehyde (PFA) for 15 min at room temperature. Cells were washed and resuspended in water. They were then mounted into slides using Vectashield Antifade Mounting Medium (Vector Labs, Newark, CA, USA) with coverslips. Cells were visualized on an inverted/DIC Zeiss Axiovert 200M microscope with an AxioCam HRm camera (Zeiss, Oberkochen, Germany), as previously described [26]. GFP imaging was performed using an excitation wavelength of 470 ± 20 nm and an emission wavelength of 525 ± 25 nm. Z stacks were analyzed and images were deconvoluted with the fast iterative method using AxioVision 4.8 software (V 4.8, Zeiss, Oberkochen, Germany).

### 2.8. Replicative Life Span (RLS)

The RLS was determined by microdissection, as outlined elsewhere [22]. Briefly, 20–30 naïve cells were isolated and arrayed in a straight line in SM plates. Every time a mother cell budded (1–2 h), the daughter cell was separated using a 25 µm needle (CoraStyles, Talent, OR, USA) under a tetrad dissection Axioscope A1 microscope (Zeiss, Oberkochen, Germany) at 250× magnification. After each budding event, the plates were incubated at 37 °C. The RLS was determined by the number of times the mother cell buds before dying (24 h without a budding event).

### 2.9. Galleria Mellonella Infection

Galleria mellonella infection was performed as previously described [27]. *G. mellonella* larvae were obtained from Vanderhorst Wholesale Inc. (St. Mary’s, OH, USA). *C. neoformans* cells were washed and diluted to 10^6^ cells/mL in PBS. The worms were injected with 10 μL of the cell suspension and PBS was used as a negative control. Twenty worms were used for each group. Survival of the worms was observed for a week. Retention of *C. neoformans* cells in the hemolymph of Galleria larvae was analyzed as an independent experiment. The worms were injected with 5 × 10^4^
*C. neoformans* cells and the hemolymph was extracted after 24 h. The samples were plated in YPD agar plates and the CFU was counted after 48 h incubation at 37 °C.

### 2.10. Growth Curve

Growth curves for H99, Δ*afr3*, Δ*afr1*, and Δ*afr2* strains were performed in 96-well flat-bottom plates, in which 0.1 OD_600_ cells were used in triplicate for each strain. The growth curve was carried on for 72 h in a SpectraMax i3x (Molecular Devices, San Jose, CA, USA) at 37 °C with shaking.

### 2.11. Expression Analysis

Strains were grown overnight in their respective media. For pump compensation analysis, H99, Δ*afr3*, and Δ*afr1* were grown overnight in SM media. *AFR3* expression in low glucose was performed in H99 cells grown overnight in SM and calorie restriction low glucose media. Finally, for pump analysis under FLC treatment, we grew H99 overnight in SM media, followed by a 2 h treatment under 32 μg/mL of FLC of 10^7^ cells. For 10-generation *C. neoformans* quantification, we isolated young and old H99 cells, where we quantified *AFR1*, *AFR2*, and *MDR1* expression. RNA was extracted using the RNAeasy Plus kit (Qiagen, Hilden, Germany), following the manufacturer’s guidelines. Next, RNA was quantified using a Biospectrophotometer (Eppendorf, Hamburg, Germany), in which an absorbance ratio (A260/A280) of 2.0 or higher was considered good quality RNA. Then, 250 ng of RNA was converted to cDNA using Verso cDNA Kit (Thermo Fisher Scientific, Waltham, MA, USA) in a 20 μL reaction. cDNA was diluted 1:5 with RNase/DNase-free water (HyClone Laboratories, Logan, UT, USA). qPCR expression analysis (Roche, Basel, Switzerland) was performed using Power Sybr Green Master Mix (Applied Biosystems, Waltham, MA, USA) following the manufacturer’s protocol. The oligonucleotides used to analyze gene expression of *AFR1* and *AFR3* are described in Appendix A. House-keeping gene *ACT1* was used as an internal control. Data were normalized and calculated using the 2^−^^ΔΔCT^ method, as previously described [28].

### 2.12. Macrophage-Mediated Killing Assay

Macrophage-mediated killing assay was performed according to the previously published protocol [18]. Briefly, 5 × 10^4^ cells of J774A.1 murine macrophage cell line were seeded in 96 well plates (Corning Costar, Corning, NY, USA) in DMEM (Gibco, Life Technologies, Carlsbad, CA, USA) media containing 10% fetal bovine serum (FBS), 10% NCTC (Gibco, Life Technologies, Carlsbad, CA, USA), 1% non-essential amino acids, and 1% penicillin-streptomycin. The 96 well plates were incubated at 37 °C with 5% CO_2_ for 24 h. After incubation, the cells were activated with LPS and IFN*γ* as described previously. In a separate tube, young and old *C. neoformans* cells were opsonized for 30 min at 37 °C with 18b7 antibody (Sigma-Aldrich, St. Louis, MO, USA), which binds to the *C. neoformans* capsule. The opsonized *C. neoformans* cells were then added to the 96 well plates containing the activated macrophages at an MOI of 1:1. The plates were incubated for 1 h at 37 °C with 5% CO_2_ to allow phagocytosis. After phagocytosis, all wells were washed 3 times with 1× PBS to remove the non-phagocytosed *C. neoformans* cells. After washing, half of the wells of macrophages were lysed using sterile water, and *C. neoformans* cells were plated in YPD to determine the number of *C. neoformans* cells phagocytosed (time 0). Next, to the other half of macrophage-containing wells, fresh DMEM media was added. The macrophages along with the phagocytosed *C. neoformans* cells were incubated for another 1 h at 37 °C with 5% CO_2_. This was carried out to analyze macrophage-mediated killing of young and old *C. neoformans* cells. After 1 h of killing, the wells were washed 3 times with 1× PBS. The macrophages were then lysed and the surviving *C. neoformans* cells were plated in YPD agar plates. YPD plates were then incubated at 37 °C for 48 h and CFUs were counted. The assay was performed in triplicate. Macrophage-mediated killing was calculated as follows:$$\%\;\mathrm{macrophage-mediated}\;\mathrm{killing}=\frac{CFU\;post\;phagocytosis\;at\;time\;0-CFU\;after\;1\;h\;killing}{CFU\;post\;phagocytosis\;at\;time\;0}\times 100\;$$

### 2.13. Statistics

Statistical analyses were performed using GraphPad Prism 9.0 (GraphPad, San Diego, CA, USA). The specific analyses are described in the figure legends. The coefficient of variation (CV) is calculated by dividing de standard deviation (SD) by the mean of the strain (CV% = SD/mean × 100).

## 3. Results

### 3.1. Afr3 Is Similar to Other ABC Transporters

Based on the amino acid sequence, *CNAG_06909* exhibits 29% and 26% identity to Afr1 (*CNAG_00730*) and Afr2 (*CNAG_00869*), respectively, with a high query coverage of above 85% (Table 1). Protein sequence identity between Afr1 and Afr2 is 38%, which is comparable to the identity of the *CNAG_06909* transporter with either Afr1 or Afr2. Furthermore, *CNAG_06909* contains an ABC transporter domain, while Afr1 contains two domains. Afr2, similarly to Afr1, also contains two ABC transporter domains. Performing an alignment and analysis employing the NCBI Blast Tool exhibits high similarity between the ABC transporter domain of Afr3 (CnAFR3) and the two domains of Afr1 (CNAFR1.1 and CnAFR1.2; 46% positives, 25% identities, and 53% positives, 35% identities, respectively) (Figure 1). Based on the similarity with Afr1 and Afr2, and the below characterized functional overlap, we have renamed the *CNAG_06909* gene *AFR3* and the protein Afr3.

### 3.2. The Afr3 Efflux Pump Is Important for C. neoformans FLC Tolerance

Given the known role of ABC transporters in mediating azole resistance in fungal cells, we assessed FLC susceptibility in H99 and Δ*afr3* with standard methods with slight modifications. First, we performed a FLC E-Test strip analysis on SM media plates, in which Δ*afr3* displayed a lower MIC than H99 (0.75 µg/mL vs. 4 µg/mL) (Figure 2A). It is noteworthy that the Δ*afr3* mutant lacked heteroresistance, whereas individual colonies grew at higher MIC in the wild-type H99, consistent with heteroresistance. Next, we explored whether *AFR3* was overexpressed during FLC treatment. After a 2h FLC treatment, *AFR3* did not show an increase in expression when compared to H99 cells without treatment (Figure 2B).

Given that augmented ABC transporter-mediated efflux of FLC is the main mechanism of how ABC pumps mediate FLC resistance in *C. neoformans*, we explored efflux pump activity of Δ*afr3*. The fluorescent dye Rhodamine 6G was added to *C. neoformans* cells and the cellular efflux was initiated by the addition of glucose, followed by fluorescence measurement in the supernatant. These data demonstrated that Δ*afr3* exhibited lower efflux than the wild-type in all three time points (10, 20, and 30 min; *p* < 0.001) (Figure 2C), indicating that Afr3 is a relevant efflux pump.

FLC MIC analysis under low glucose media (0.05% glucose) was performed because such conditions are encountered by *C. neoformans* in vivo. These data showed that both strains increased FLC tolerance under low glucose, with no difference between the H99 and Δ*afr3* strains, indicating that Afr3 is not responsible for the increase in FLC tolerance observed under low glucose conditions (Figure 2D). Of note, the FLC MIC_80_ was the same for H99 and Δ*afr3* in a broth microdilution assay. These data were further supported by the expression analysis of *AFR3* under low glucose, in which the *AFR3* levels were not significantly increased (Figure 2E). Possible pump compensation between Afr3 and Afr1 was also studied. *AFR1* expression was measured in the Δ*afr3* strain, while *AFR3* expression was measured in the Δ*afr1* strain to test for upregulation. Upregulation of *AFR1* in an Δ*afr3* strain would indicate that there is increased production of Afr1 to compensate for the lack of Afr3, and vice-versa. No upregulation was observed in this analysis, which indicates no compensation between Afr1 and Afr3 (Appendix A).

Given that Afr3 (fold-change = 16.06; *p*-value = 0.0017), as well as Afr1 and Afr2, are overexpressed in 10-generation old cells (Table 2) [15], we assessed the contribution of Afr3 to age-associated FLC tolerance with established high dose FLC killing assays, since growth-based assays can only assess FLC MICs of young *C. neoformans* cells. Killing of young (Y) and old (O) *C. neoformans* cells was assessed after 3 h exposure to high doses of FLC ranging from 50 µg/mL to 3.125 µg/mL. As expected, these assays demonstrated that the Δ*afr3* mutant was killed more efficiently over a range of FLC doses when compared to the wild-type H99 (Δ*afr3* Y vs. H99 Y, FLC 50, 25, 12.5 µg/mL: 80 vs. 25%; FLC 6.25 µg/mL: 48 vs. 7%; FLC 3.125 µg/mL: 60 vs. 5%; ### *p* < 0.001) (Figure 2F). Older *C. neoformans* cells showed higher tolerance to FLC killing than younger *C. neoformans* cells for most FLC concentrations for both H99 (H99 O vs. H99 Y; FLC 50 µg/mL: 13 vs. 25%, *p* < 0.05; FLC 25 µg/mL: ~0 vs. 23%, *p* < 0.01; and FLC 12.5 µg/mL: 6 vs. 25%, *p* < 0.01) and Δ*afr3* (Δ*afr3* O vs. Δ*afr3* Y; FLC 50 µg/mL: 17 vs. 80%; FLC 25 µg/mL: 35 vs. 80%; and FLC 12.5 µg/mL: 37 vs. 79%; *p* < 0.01). The H99 wild-type data corroborate the data previously published for the RC2 strain [19]. The percentage FLC killing of older Δ*afr3* cells, however, was higher than the killing of 10-generation H99 cells (Δ*afr3* O vs. H99 O; FLC 25 µg/mL: 35 vs. ~0%, FLC 12.5 µg/mL: 37 vs. 6%; and FLC 6.25 µg/mL: 38 vs. ~0%; $ *p* < 0.05), indicating that the presence of the Afr3 efflux pump may contribute to age-associated FLC tolerance, but is not the only factor.

### 3.3. S. cerevisiae Expression of Afr3 Increases Drug Efflux and Resistance

To exclude compensation by other ABC transporters when the Afr3 pump is deleted, we expressed Afr3 in the *S. cerevisiae* ADΔ strain, which lacks all seven of the main ABC transporters, thus reducing background transport. We expressed Afr3 in the pYES2 plasmid under a *GAL1* promoter (Figure 3A) that permits the expression of Afr3 only in the presence of galactose. The ADΔ cells expressing Afr3 were called ADΔ + Afr3, while ADΔ transformed with an empty vector was employed as the control.

Since Afr3 is an ABC transporter, we sought to evaluate the efflux activity of the *S. cerevisiae* strains at 30 °C. First, we performed a Rhodamine 6G efflux assay, in which the efflux was increased in the ADΔ + Afr3 strain when compared to ADΔ after 30 min (*p* < 0.001) (Figure 3B). Similarly, a Nile Red Assay was performed, in which cells were incubated with Nile Red, and glucose was added to initiate transport. The percentage of Nile Red accumulation inside the fungal cells was quantified by measuring the fluorescence after 30 min. ADΔ + Afr3 exhibited decreased intracellular Nile Red accumulation when compared to the ADΔ control strain (73% vs. 100%; *p* < 0.01) (Figure 3C).

Since the expression of plasma membrane proteins can alter membrane permeability, we constructed an Afr3 mutant protein that lacks the nucleotide-binding domain (NBD). This mutant is catalytically dead, since ATP cannot bind to it due to the absence of NBD. The ADΔ cells expressing the NBD mutant did not show a statistical difference in Nile Red accumulation from the ADΔ control at 30 °C. However, the Nile Red accumulation in the NBD mutant significantly differed from the accumulation in the ADΔ + Afr3 strain (*p* < 0.05) (Figure 3C). This demonstrates that the effect observed by Afr3 expression is not due to membrane alterations.

Lastly, we tested the effects of Afr3 expression on drug resistance. We performed an FLC E-Test strip analysis on galactose media plates at 30 °C, in which the expression of Afr3 rendered the cells more resistant to FLC (MIC = 0.064 µg/mL) in comparison to the ADΔ control, which showed an MIC marginally lower than the detectable threshold of the assay (MIC < 0.016 µg/mL) (Figure 3D). The present data confirm the findings in *C. neoformans*, in which Afr3 presents relevant activity as an efflux pump that contributes to FLC resistance.

### 3.4. Afr3 Localizes at the Cell Surface

To further understand the function of Afr3, we introduced a C-terminally tagged *AFR3*-GFP cassette into the *S. cerevisiae* ADΔ cells using the pYES2 plasmid. Fluorescence levels were assessed using an inverted/DIC Zeiss Axiovert microscope on both strains under the same exposition and superposed to brightfield images. Our results demonstrated that GFP fluorescence is higher in the ADΔ + *AFR3*-GFP cells when compared to the ADΔ control (Figure 3E). This suggests that the Afr3 protein is expressed in ADΔ, consistent with the *AFR3* expression data of ADΔ and ADΔ + Afr3 cells, which demonstrate a 425-fold increase in expression (Figure 3F).

The ADΔ + *AFR3*-GFP strain also allows for the visualization of Afr3 localization within the fungal cell. Z stack images reveal fluorescence at the cell surface, suggesting localization in the plasma membrane, which is consistent with the Afr3 efflux pump function (Figure 4). White arrows indicate regions with surface localization of Afr3, which were universally distributed and focal in other cells. In addition to the surface localization, images revealed punctate and network-like localization of Afr3 within the cytoplasm. Although it is not clear if this cytoplasmic localization is correlated to a specific organelle, this sub localization could indicate a potential secondary function of Afr3.

### 3.5. Afr3 Affects Cryptococcal Virulence

Next, we assessed if Afr3 plays a role in virulence. First, we documented that the mutant strain exhibited no growth defect at 37 °C when compared to the wild type (Appendix A). Next, we analyzed phagocytosis and macrophage killing of Δ*afr3* and compared the percentages to the H99 wild type. These data indicated that the phagocytosis of Δ*afr3* was comparable to that of H99. However, murine J774 macrophages more successfully killed the Δ*afr3* mutant strain after phagocytosis when compared to H99 wild-type (64% vs. 6.6%, *p* < 0.01) (Figure 5A). Last, the virulence of the mutant strain was assessed in a *G. mellonella* infection model. The larvae infected with the Δ*afr3* strain exhibited increased survival when compared to the larvae infected with H99 (8 vs. 6 d median survival, *p* < 0.0001), indicating that the Afr3 pump plays a role in virulence (Figure 5B). These data are further supported by the lower number of *C. neoformans* cells circulating in the hemolymph of larvae infected with Δ*afr3* after 24 h when compared to CFU from the hemolymph of larvae infected with H99 (29 vs. 52% cells retained, *p* < 0.05) (Figure 5C). These data support the notion that Afr3 is also important for *C. neoformans* virulence.

### 3.6. Afr3 Affects Cryptococcal Replicative Life Span

Given that Afr3, as well as Afr1 and Afr2, are overexpressed in 10-generation old cells, we first investigated whether Afr1, Afr2, and Afr3 affect RLS. The RLS of Δ*afr3*, Δ*afr1*, and Δ*afr2* were determined by micro-dissection and the total number of divisions the respective mutant and wild-type *C. neoformans* cells undergo before their death is recorded. These experiments showed that loss of AFR3 had a moderate prolongevity effect and extended the median RLS by 29% compared to the wild-type H99 strain (22 vs. 17, *p* < 0.001) (Figure 6). In contrast, loss of AFR1 and AFR2 did not alter the RLS (15 and 17 median RLS, respectively). The RLS of Δ*afr1* and Δ*afr2* mutants exhibited high variability, whereas RLS of Δ*afr3* was characterized by lower stochasticity. The coefficient of variation measures the amount of variation between individual cells within the strain data set. H99 showed 35% of variation between individual cells, while Δ*afr3* had only 14% variation, in accordance with its lower stochasticity. The mutant strains Δ*afr1* and Δ*afr2* had a much higher variation, with 55% and 65%, respectively.

## 4. Discussion

This paper describes a novel ABC transporter in *C. neoformans*, which can efficiently efflux Rhodamine 6G and Nile Red, two fluorescent dyes used to determine the function of efflux pumps, and render *C. neoformans* cells more resistant to FLC. The present study was initiated because in 10-generation-old cells [15], which are more tolerant to FLC, the *AFR3* gene is markedly upregulated, similarly to the upregulation of *AFR1* and *AFR2* genes. Based on the similarities with known *C. neoformans* efflux pumps, Afr1 (*CNAG_00730*, 29.29% identity), Afr2 (*CNAG_00869*, 26% identity), and Pmr5 (*CNAG_06348*, 25% identity), as well as ABC transporters from the environmental fungi *Ustilago trichophora* (41.35% identity), *Lasallia pustulata* (54.27% identity), and *Saitozyma podzolica* (73.22% identity), these data characterize Afr3 as member of the conserved family of ABC transporters.

Overexpression of efflux pumps, leading to decreased cellular drug concentration, is a major mechanism of drug resistance [29]. During treatment of chronic cryptococcosis, this can lead to persistence of infection, despite appropriate treatment, which translates into failure to clear the fungal cells [30]. Deletion of *AFR3* resulted in increased sensitivity to FLC in the E-Test strip analysis and increased CFU killing when higher FLC concentrations were used. However, Δ*afr3* did not exhibit differences in FLC sensitivity using the standard MIC_80_ assay. E-Test and microdilution methods do not always yield the same results [31], and agreement between these two tests can vary 70–96% of the time [32]. It is conceivable that the observed difference could be due to “trailing growth”, a phenomenon where fungal cells exhibit reduced but persistent growth at FLC concentrations above the MIC. This effect is more commonly observed when microdilution methods are performed [33] because in liquid assays, slow growth of subpopulations may eventually dominate [34]. Interestingly, the E-Test assay indicated a lower formation of heteroresistant colonies in the Δ*afr3* mutant strain when compared to H99. Heteroresistance signifies the presence of a sub-population that manifests higher FLC tolerance when compared to the majority of the population [35,36]. Loss of heteroresistance was also observed with the deletion of *AFR1* [14]. However, the role that Afr3 plays in heterotolerance mechanisms is still to be determined.

Furthermore, the decreased Rhodamine 6G efflux, which mimics alterations in drug accumulation, corroborates the function of Afr3 as an efflux pump. Transcription data also suggest that there is no compensation by the other main efflux pump Afr1. Afr1 is the most well-characterized ABC transporter in *C. neoformans.* Deletion of *AFR1* increases drug susceptibility to FLC, which is substantiated by data from a mouse infection model that indicates that Afr1 also plays a role in FLC resistance and fungal virulence [13,35,37]. Single deletions of Afr2 and Mdr1 did not influence susceptibility to FLC, which was only observed in a triple-deletion of Afr1, Afr2, and Mdr1 [14]. Compensatory upregulation of the transporters by qPCR in the Δ*afr3* and Δ*afr1* mutant strains was evaluated to assess if the lack of Afr3 would be compensated by Afr1. The data did not indicate overexpression of any transporter when the other was deleted.

Heterologous protein expression in yeast model systems, such as *S. cerevisiae*, has enabled functional analysis of specific proteins and has been previously employed to study efflux pumps in a variety of fungi [6,20,38]. Interestingly, Afr3 effectively exports both Rhodamine 6G and Nile Red out of the mutated *S. cerevisiae* strain, whereas the Afr1 pump only exports Nile Red and not Rhodamine 6G efflux [14]. This suggests that Afr3 and Afr1 differ in their substrate specificity. In *S. cerevisiae*, *AFR3*-GFP localized to the cell surface, as well as the cytoplasm. Based on Afr3’s six transmembrane domains, we propose that Afr3 may be expressed in membranes of organelles in the cytoplasm. ABC transporters of *Candida glabrata*, *Candida albicans*, *S. cerevisiae*, and *Aspergillus fumigatus* [37,38,39,40] have been shown to localize in the membranes of mitochondria and the vacuole [41,42]. Future studies are necessary to analyze the subcellular localization of Afr3, especially during aging, since Afr3 expression negatively affects the replicative life span.

Both drug resistance and heteroresistance have been linked to increased virulence [35]. The decreased cryptococcal virulence of Δ*afr3* in a *Galleria mellonella* survival model supports this notion. The efflux pumps may not only play a role in FLC efflux but also contribute to detoxification of the cell by extruding metabolites and other toxic components [43]. Accumulation of toxic metabolites can lead to a loss of cell fitness and impact fungal virulence. Other fungal pumps, including Atm1, a mitochondrial ABC pump in *C. neoformans*, Mlt1, a vacuolar ABC transporter in *C. albicans*, and AbcB, an efflux pump of *A. fumigatus*, have been shown to impact virulence [40,41,42]. Efflux pump-dependent detoxification has been associated with a loss of lifespan in *S. cerevisiae* [17]. Instead, we documented a small increase in RLS of the Δ*afr3* strain and a loss of lifespan stochasticity, which was not observed in Δ*afr1* and Δ*afr2.* This could indicate that Afr3 has a role in stress response.

Exposure to stress, such as glucose deprivation, leads to an increase in FLC tolerance, as the change is only transient and does not become intrinsic to the strain [26]. Low glucose conditions lead to the increased pump activity of Afr1 [26]. In contrast, *AFR3* deletion did not affect the enhanced FLC tolerance under glucose starvation, nor did we observe upregulation of *AFR3*. *C. neoformans* can achieve resistance to FLC by the duplication of chromosome 1 and rarely chromosome 3 in response to prolonged exposure to FLC. Chromosome 1 is most commonly duplicated, increasing the copy number of genes *ERG11* and *AFR1* [44]. The observed increased FLC tolerance in low glucose growth conditions, however, is not associated with a change in the gene copy number of *AFR1* or *AFR2*. [26]. *AFR3* is located on chromosome 3 and preliminary analysis show that the gene copy number was unchanged under low glucose stress [14,26]. Afr3, as well as the other characterized ABC transporters (Afr1, Afr2, and Mdr1), are upregulated in 10-generations old cells. Additionally, the results from the modified killing assay showed that deletion of *AFR3* in older cells leads to a partial loss of resistance, indicating that Afr3 could aid in the FLC tolerance observed in 10-generation cells.

In summary, these data identify a novel ABC transporter that contributes to FLC tolerance in *C. neoformans* older cells. This ABC transporter promotes drug efflux from the fungal cell, influencing the susceptibility to treatment and the virulence of the cryptococcal cells. Our data encourage further efforts to understand its role in relation to other efflux pumps and which mechanisms the aging cells employ to increase drug resistance.

## Figures and Tables

**Figure 1 jof-08-00677-f001:**
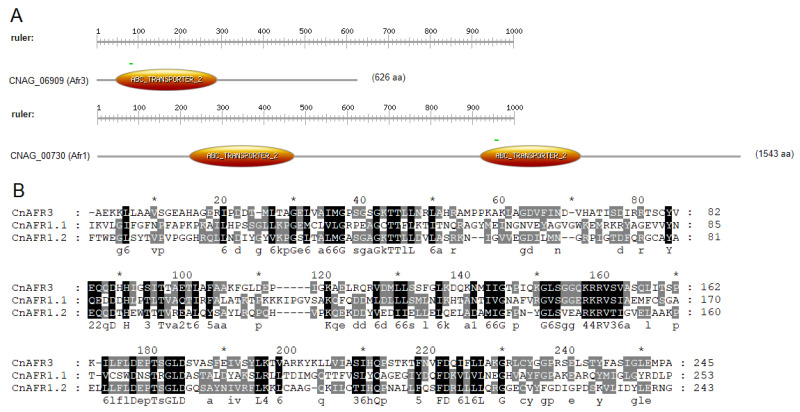
**Afr3 (*CNAG_06909*) Has Similar Motifs to Afr1 (*CNAG_00730*).** (**A**) Afr3 possesses one ATP-binding cassette (ABC) domain, an ATPase domain that utilizes ATP binding and hydrolysis to fuel the transport of different molecules across membranes, while Afr1 has two. The search was conducted using Prosite (https://prosite.expasy.org/scanprosite/; accessed on 3 December 2021); (**B**) alignment analysis between the ABC transporter domain of Afr3 (CnAFR3) and the two domains of Afr1 (CnAFR1.1 and CnAFR1.2) was performed with ClustalW, in the BioEdit program. The figure was generated using GeneDoc. Sequences in black are highly conserved.

**Figure 2 jof-08-00677-f002:**
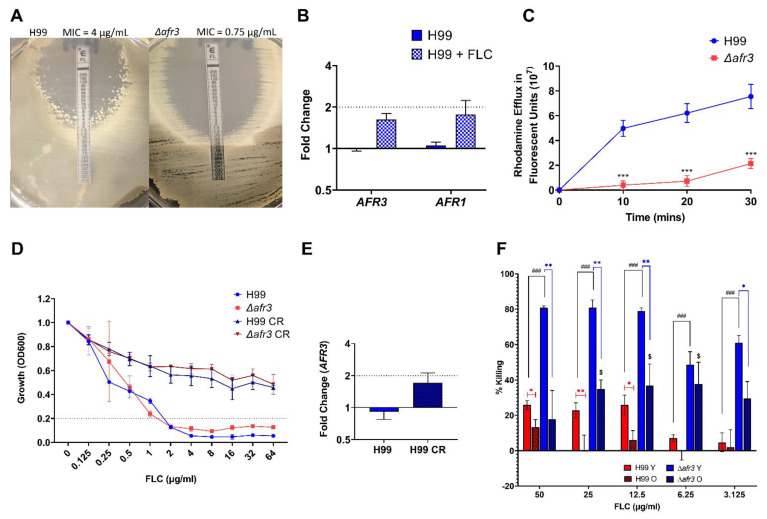
**Afr3 is an Important Pump for Drug Resistance.** (**A**) *C. neoformans* Δ*afr3* is more sensitive to FLC than H99 in an FLC E-Test in a YPD plate; (**B**) H99 cells that underwent FLC treatment with 32 μg/mL for 2 h (checkered blue/white bar) do not increase expression of *AFR3* and *AFR1* compared to H99 wild-type (blue bar); (**C**) Δ*afr3* (red line) decreases efflux compared to H99 (blue line) in a Rhodamine 6G assay. Statistical analysis was performed with multiple unpaired Student’s *t*-test, *** *p* < 0.001; (**D**) FLC tolerance observed under CR conditions (SM 0.05% glucose) (H99 CR: dark blue line, Δ*afr3* CR: dark red line) is independent of Afr3 presence, as shown by the susceptibility under normal conditions (SM 2% glucose) (H99: blue line, Δ*afr3:* red line); (**E**) expression of *AFR3* is not increased under CR conditions (checkered blue bar) when compared to normal glucose conditions (blue bar); (**F**) the Δ*afr3* young cells (Δ*afr3* Y, blue bar) are more susceptible to FLC killing than H99 young cells (H99 Y, red bar). Furthermore, Δ*afr3* old cells (Δ*afr3* O, checkered blue bar) lose FLC killing tolerance when compared to H99 old cells (H99 O, checkered red bar). Statistical analysis was performed with multiple unpaired Student’s *t*-test, * *p* < 0.05, ** *p* < 0.01, ### *p* < 0.001, and $ *p* < 0.05; error bars represent the standard deviation between biological triplicates.

**Figure 3 jof-08-00677-f003:**
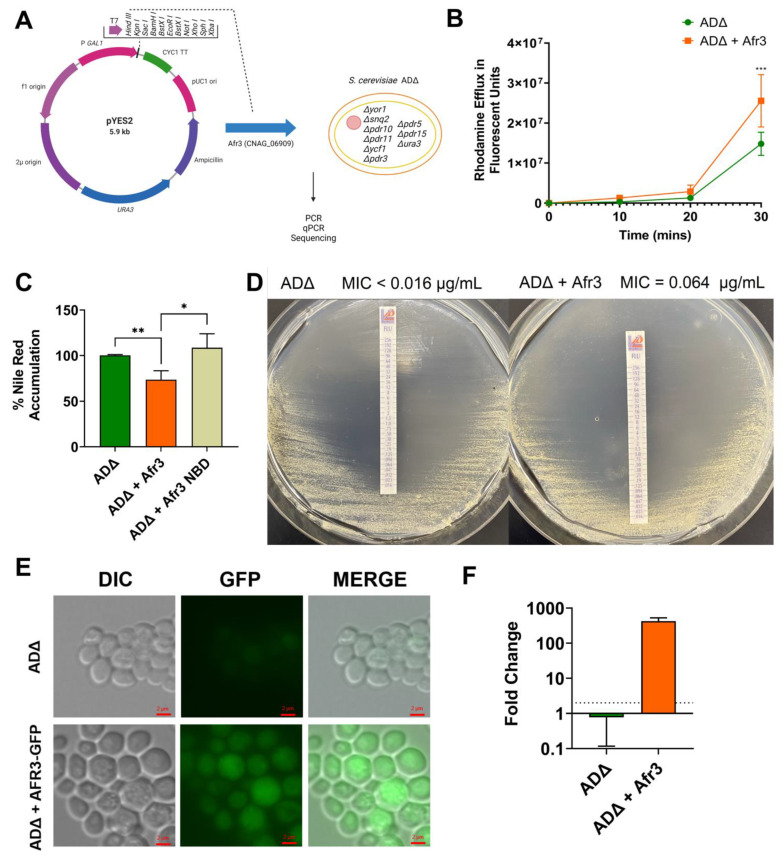
**Afr3 Expression in *Saccharomyces cerevisiae* Increases Efflux and FLC Resistance.** (**A**) Schematic diagram of Afr3 expression in *S. cerevisiae* ADΔ strain. The pYES2 plasmid contains a *URA3* auxotrophic marker gene, a 2µ origin of replication, a *GAL1* promoter, and a *CYC1* terminator. Hind III was used as the cloning site between the *GAL1* promoter and *CYC1* terminator. *AFR3* (*CNAG_06909*) cassette was inserted in this cloning site when transformed into *S. cerevisiae* ADΔ strain. The transformation was confirmed through plasmid PCR, qPCR for *AFR3* expression, and sequencing of the *AFR3* cassette. The figure was designed with BioRender. All subsequent experiments were performed at 30 °C; (**B**) ADΔ + Afr3 (orange line) shows increased efflux when compared to ADΔ (green line) when measured employing the Rhodamine 6G dye; (**C**) ADΔ + Afr3 (orange bar) has lower intracellular accumulation of Nile Red when compared to control ADΔ (green bar). A catalytically inactive NBD mutant of Afr3 (yellow bar) does not show a statistical difference in Nile Red accumulation from ADΔ; (**D**) a FLC E-Test strip assay shows that expression of Afr3 increases the strain resistance to FLC (MIC of 0.064 µg/mL vs. <0.016 µg/mL); (**E**) *S. cerevisiae* ADΔ cells expressing *AFR3*-GFP show protein expression by means of GFP fluorescence. Brightfield and epifluorescence images were obtained that were later superimposed; (**F**) *AFR3* expression in ADΔ is 425-fold higher (orange bar) than the control (green bar) as measured by qPCR. Error bars represent the standard deviation between biological triplicates. Statistical analysis was performed with multiple unpaired Student’s *t*-test, * *p* < 0.05, ** *p* < 0.01, *** *p* < 0.001.

**Figure 4 jof-08-00677-f004:**
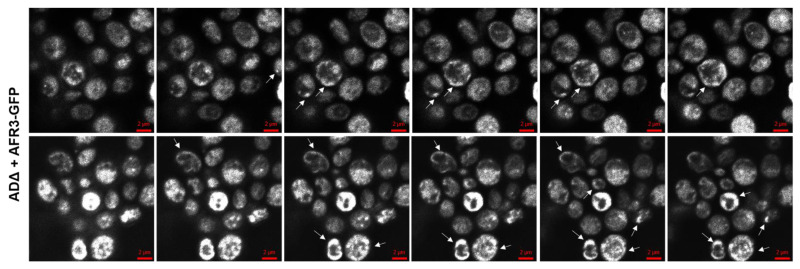
**Afr3 Localizes in the Plasma Membrane.** *S. cerevisiae* ADΔ cells expressing *AFR3*-GFP show fluorescence on the cell surface, suggesting localization at the level of the plasma membrane. White arrows point to localization within fungal cells. Images were obtained through the generation of ten Z stacks, six of which are represented in this image.

**Figure 5 jof-08-00677-f005:**
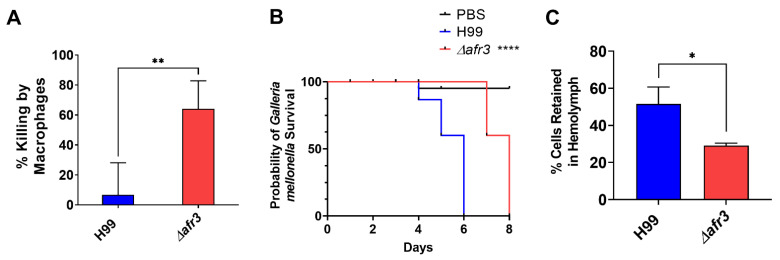
**Afr3 Plays a Role in Virulence.** (**A**) Phagocytosed Δ*afr3* (red bar) cells are better killed by J774A.1 murine macrophages than wild-type H99 cells (blue bar). Statistical analysis was performed with Student’s *t*-test, ** *p* < 0.01; (**B**) *Galleria mellonella* larvae survived longer when infected with mutant strain Δ*afr3* (red line), when compared to the survival of the larvae infected with wild-type H99 (blue line). Black line represents PBS uninfected controls. Statistical analysis was performed with log-rank (Mantel–Cox) test, **** *p* < 0.0001; (**C**) Δ*afr3* (red bar) has lower retention of *C. neoformans* cells in the larvae hemolymph than H99 (blue bar). Statistical analysis was performed with Student’s *t*-test, * *p* < 0.05; error bars signify standard deviations between the biological triplicate.

**Figure 6 jof-08-00677-f006:**
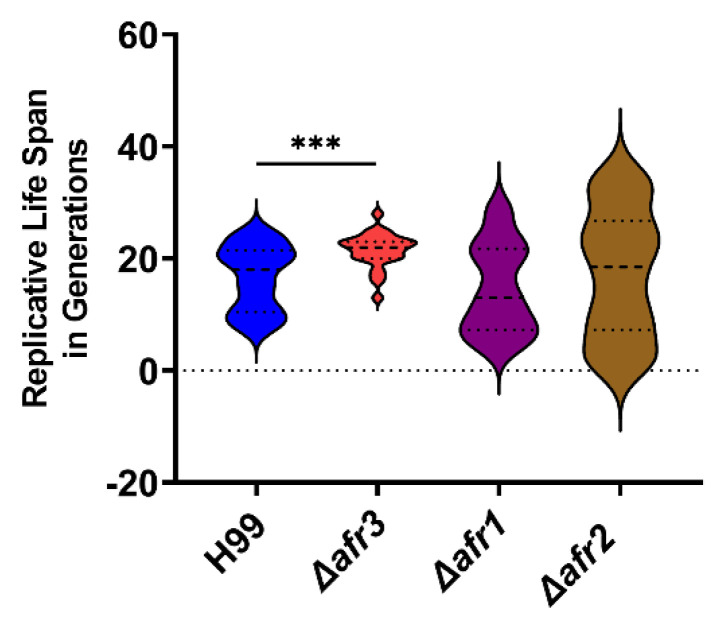
**Afr3 Plays a Role in Aging.** The *C. neoformans* Δ*afr3* mutant strain (red) has an expansion of the median lifespan compared to the H99 wild-type strain (blue), while Δ*afr1* (purple) and Δ*afr2* (brown) mutant strains do not show significant difference. Statistical analysis was performed with Student’s *t*-test, *** *p* < 0.001.

**Table 1 jof-08-00677-t001:** Similar Proteins to Afr3 across Different Fungal Species.

Description	Max Score	Total Score	Query Cover	E Value	% Identity	Protein Length	Accession Code
ATP-binding cassette transporter (Cryptococcus neoformans var. grubii H99)	205	302	86%	3 × 10^−55^	29.29%	1543	*CNAG_00730*
ABC transporter (Cryptococcus neoformans var. grubii H99)	199	340	88%	3 × 10^−53^	31%	1462	*CNAG_07799*
ATP-dependent permease (Cryptococcus neoformans var. grubii H99)	179	234	77%	1 × 10^−44^	36.96%	1173	*CNAG_06533*
ABC transporter family protein (Cryptococcus neoformans var. grubii H99)	170	313	61%	9 × 10^−44^	30.99%	1241	*CNAG_05470*
ABC transporter PMR5 (Cryptococcus neoformans var. grubii H99)	165	315	87%	5 × 10^−42^	25.05%	1421	*CNAG_06348*
ATP-binding cassette transporter (Cryptococcus neoformans var. grubii H99)	144	271	90%	1 × 10^−39^	26.06%	1529	*CNAG_00869*
ABC transporter, putative (Cryptococcus neoformans var. neoformans JEC21)	204	343	86%	3 × 10^−56^	30.16%	1463	*CNL06490*
ABC transporter (Cryptococcus gattii VGIV IND107)	1167	1167	100%	0.0	93.45%	625	*KIR83217.1*
related to ATP-binding cassette protein (ABC) transporter (Ustilago trichophora)	208	413	85%	3 × 10^−58^	41.35%	669	*SPO26649.1*
ABC transporter (Lasallia pustulata)	671	671	96%	0.0	54.27%	637	*KAA6412381.1*
Hypothetical protein EHS25_004009 (Saitozyma podzolica)	910	910	95%	0.0	73.22%	1711	*RSH94206.1*

**Table 2 jof-08-00677-t002:** Efflux Pumps Expression in 10-generation Cells.

Strain	qPCR Fold-Change	*p* Value	Significant?
Afr1	28.26	0.0415	Yes
Afr2	5.16	0.0026	Yes
Mdr1	4.36	0.0108	Yes

## Data Availability

All data required to understand this article are presented in the study or the Appendix A. Any raw data further requested will be provided from the corresponding author.

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
