# Peer review of "Novel ABC Transporter Associated with Fluconazole Resistance in Aging of Cryptococcus neoformans"

_jof, 2022, doi:10.3390/jof8070677_

Round 1
Reviewer 1 Report
In my opinion, I would accept this work for publication.
Author Response
Thank you for reviewing our paper.
Reviewer 2 Report
The manuscript ‘Novel ABC Transporter Associated with Fluconazole Resistance in Aging of Cryptococcus neoformans’, corresponds to an updated version of a manuscript previously submitted to Journal of Fungi. From the methodical point of view I have no complaints. In this submission the authors have addressed all recommendations to the previous version of the manuscript.
Overall, the manuscript now is well written, and in my opinion is worthy to publish in Journal of Fungi.
I have some minor remarks, such as:
row 142: employing oligonucleotides
row 147: using oligonucleotides
row 148: The oligonucleotides
Author Response
Thank you for reviewing our paper. The minor remarks have been corrected as suggested.
Reviewer 3 Report
The authors perform an interesting study detecting a new ABC transporter associated with fluconazole resistance in aged C. neoformans cells. The work carried out is quite complete and its conclusions are interesting.
There are a few things to be clarified, which I will now summarize as follows:
- Mutant strains without this AFR3 gene appear to have partially deregulated growth and division. Since deletion of the other AFR genes, AFR1 and AFR2, are also observed to show growth deregulation. Can the authors explain how these transporters are related to the growth inhibition they appear to cause with their presence or why they are dysregulated in their absence? In addition, these deletion mutants appear to be more sensitive to FLC. It is known that exponentially growing cells are more sensitive to antimicrobial treatments and that this dysregulation may affect growth, perhaps some conclusions may be biased.
- There is one comment I would like to make to the authors: On the one hand, this new Afr3 transporter seems to be overexpressed in aged cells (which reach the limit of budding that mother cells are able to perform, which could imply a slower growth). On the other hand it seems that the presence of this ABC transporter is protecting the fungus from the antifungal fluconazole. Moreover, the authors, when using a low concentration of glucose that would cause a very slow growth or no growth of the fungus, see that tolerance in all strains to FLC increases but that this transporter does not seem to be responsible for it. If host glucose concentration is low during infection, would this overexpression of the AFR3 gene actually protect the fungus from fluconazole treatment? Could it be that in part the possible higher FLC resistance observed in aged cells is due in part to that lower or less rapid growth (which would not need such an active ergosterol metabolism) and not only to Afr3 transporter activity causing FLC output? And the same in the resistance to macrophage activity? Could it be that the transporter causes the output of toxic, oxidative, ... components and not so much to FLC output? I think it should be better clarified in the discussion.
- The authors use RNA from the fungus to produce cDNA, and then amplify the AFR3 gene by PCR from that cDNA and then bind it to the plasmid pYES2 to perform the transformation. This is perhaps a bit complicated initially, is there an ulterior motive for not using genomic DNA as a template for amplification? Do the genes you are studying have too many introns, would there be expression difficulties in Saccharomyces cerevisiae species? They should explain this choice a bit, perhaps in discussion.
- One doubt I have is that, as far as I can see, the binding of the plasmid and the amplified AFR3 cassette/amplified is achieved in vivo, inside the ADΔ strain. is this so? I think it would have been easier to bind it in vitro before performing the transformation of the strain and ensure the binding at the correct position of this gene as it is not easy when using only one restriction enzyme.
- In the killing test, the authors to opsonize Cryptococcus cells use the 18b7 antibody. Is this antibody commercial or self-produced? It should be indicated. They should also explain its target if known.
- In the legend of Figure 2B it is stated that "AFR3 and AFR1 expression is not increased", maybe the authors do not consider it overexpression because it is not greater than the 2-fold increase threshold, but I think that an increase is observed, isn't it? Even if a statistical analysis is applied I would be surprised if the expression is not statistically significant between them. If they prolong the FLC treatment for more than 2 h, this difference in expression may widen.
- In the same legend (Figure 2E) the authors again state that "AFR3 expression is not increased under CR conditions", but what is observed is an increase, although in this case it is not significant or not overexpressed.
- Is Figure 3F necessary? It is expected that the non-transformed Saccharomyces strain with the AFR3 gene shows no expression of this gene, and indicating in the legend that in the cells that have been transformed there is 425 times more expression than in the non-transformed control would not require a figure and can be included in the text.
- On the other hand, in the legends there is a repetition that I believe is unnecessary of the statistics used, sometimes the same ones, for each of the parts of the figures where they are applied. To collect at the end of each legend a single sentence with the statistics applied and the symbols used to show that they are significant would be sufficient in my opinion.
- The title of the section "3.5 Afr3 Affects Cryptococcal Life Span" should be "3.5 Afr3 Affects Cryptococcal Replicative Life Span". It does not affect the entire life span but the replicative life span and that is what is being studied, isn't it?
Minor changes:
- In lines 300-302, when comparing the ABC domains of the genes with Blast, they indicate the % identity, but in this case they also include a % of positives. What do they mean? Similarity?
- Lines 344-361: The authors start using abbreviations for young and old cells as Y and O that should be defined where these young and old terms first appear (line 347) to improve understanding.
- Figure 2 B and E do not use the same scale for Fold Change. Unify.
- Line 371: A parenthesis must be closed.
